# Long-Term Impact of the Great Chinese Famine on the Risks of Specific Arrhythmias and Severe Hypertension in the Offspring at an Early Stage of Aging

**DOI:** 10.3390/jpm13020163

**Published:** 2023-01-17

**Authors:** Qiutong Zheng, Jianhong Pu, Can Rui, Yumeng Zhang, Na Li, Yun He, Ying Gu, Yang Ye, Xiuwen Zhou, Zhice Xu

**Affiliations:** 1Maternal and Child Health Care Hospital of Wuxi & First Affiliated Hospital of Soochow University, Suzhou 215006, China; 2Nanjing Maternity and Child Health Care Hospital, Nanjing 210004, China

**Keywords:** famine, hypertension, bradycardia, atrial fibrillation, atrioventricular block

## Abstract

Perinatal malnutrition affects postnatal cardiovascular functions. This study used the Great Chinese Famine (GCF) to determine the long-term impact of perinatal undernutrition on hypertension and arrhythmias in older offspring. Subjects (*n* = 10,065) were divided into an exposed group whose fetal life was in the GCF and an unexposed group. The exposed group showed higher systolic/diastolic pressure, heart rate, and total cholesterol. Perinatal exposure to the GCF was a significant risk to Grade 2 and Grade 3 hypertension (OR = 1.724, 95%CI: 1.441–2.064, *p* < 0.001; OR = 1.480, 95%CI: 1.050–2.086, *p* < 0.05) compared to the control. The GCF also increased risks for myocardial ischemia (OR = 1.301, 95%CI: 1.135–1.490, *p* < 0.001), bradycardia (OR = 1.383, 95%CI: 1.154–1.657, *p* < 0.001), atrial fibrillation (OR = 1.931, 95%CI: 1.033–3.610, *p* < 0.05), and atrioventricular block (OR = 1.333, 95%CI: 1.034–1.719, *p* < 0.05). Total cholesterol, diabetes, and metabolic syndrome were associated with Grade 2 or Grade 3 hypertension after exposure to the GCF; high cholesterol, high BMI, diabetes, metabolic syndrome, and elevated blood pressure were linked to certain types of arrhythmias in exposed offspring. The results first demonstrated perinatal undernutrition was a significant risk factor for the development of Grade 2–3 hypertension and certain arrhythmias in humans. Perinatal undernutrition still significantly impacted cardiovascular systems of the aged offspring even 50 years after the GCF. The results also provided information to a specific population with a history of prenatal undernutrition for early prevention against cardiovascular diseases before aging.

## 1. Introduction

Cardiovascular diseases (CVD) are a major global health problem. A growing body of evidence suggests a convincing relationship between perinatal malnutrition and increased risks of CVD later in life. Laboratory studies in animal models demonstrated that maternal undernutrition during pregnancy could significantly affect heart functions in the offspring [1]. Although cardiovascular functions could be influenced by aging [2,3], less is known about the effects of perinatal undernutrition on certain specific cardiovascular diseases such as pre-excitation syndrome, atrial fibrillation, and atrioventricular block in the old offspring of humans. In this regard, the Great Chinese Famine (GCF) occurred 50 years ago, provides an interesting model to study the influence of perinatal undernutrition on cardiovascular systems in older human offspring.

The Great Chinese Famine occurred between 1959 and 1961 due to extremely severe food shortage, and it affected whole regions and populations in China, with thousands of people dying from hunger. It is well known that the people in the GCF, including pregnant women, suffered severe undernutrition. Our report several years ago was the first to show blood pressure in the offspring at middle to old age was significantly affected by the GCF [4]. Since then, several studies have reported that exposure to the GCF is an important risk factor for inducing hypertension in the offspring [5,6,7,8]. In clinical practice, hypertension is divided into three stages: Grades 1, 2, and 3 hypertension. Importantly, Grade 2 and Grade 3 hypertension have been shown to be closely related to myocardial ischemia [9]. Notably, there has been very limited information about the influence of famine or perinatal undernutrition on Grade 1, 2, and 3 hypertension and various arrhythmias in humans.

This study also aimed to examine the impact of the GCF on various arrhythmias, including bradycardia, tachycardia, premature beat, pre-excitation syndrome, atrial fibrillation, and atrioventricular block, in the old offspring, since there is very limited data available regarding the influence of prenatal undernutrition on multiple types of arrhythmias in the old offspring of humans. Interestingly, if risk factors can be identified early, certain CVDs could be prevented and/or slowed down by early interventions. In the context of personalized medicine, if certain perinatal factors and risks of developing hypertension or arrhythmias could be confirmed by investigations, personalized treatments or prevention measures can be considered for those specific offspring with a history of exposure to the certain perinatal factors. The new data gained could be a critical contribution to reducing the risks of CVDs in developmental origins. Therefore, the present study investigated the long-term influence of the GCF on the risks of Grade 1–3 hypertension and various arrhythmias in the offspring between 50 and 60 years old following exposure to perinatal undernutrition such as the GCF half a century ago.

## 2. Materials and Methods

### 2.1. Subjects

This study was performed mainly at the First Hospital of Soochow University. The data for this study were gathered from total of 34,347 people who went to the hospital for medical examinations during the selected time periods. The patients and their data were from the First Hospital of Soochow University, which is the largest hospital in the area, and patients were from the whole region in the Suzhou area. For the exposed group, the subjects’ data were collected during 2013–2018. After excluding age-inappropriate persons, excluding duplicate medical examination cases, and excluding smokers and alcohol drinkers, there were a total of 10,065 subjects in this study. A total of 6128 subjects were exposed to the GCF during fetal stages, while 3937 subjects were not exposed to the GCF. People who were born in 1959–1961 was defined as those fetal stages must be within GCF period. For non-exposure group, all subjects’ fetal stages must be after GCF, born during 1963–1965. Considering there were a few years of differences in ages between the two groups, the non-exposure group’s physical examinations in this study were a few years later than those of the exposure group, so that age differences were balanced between the two groups. All subjects gave their informed consent for inclusion before they participated in the study. The study was conducted in accordance with the principles of the Declaration of Helsinki, and all protocols were approved by the Institute Ethics Committee of the First Hospital of Soochow University (ER0201601197).

### 2.2. Measurements

People who were born in 1959 and 1961 were classified based on fetal stages falling within the GCF period. For the non-exposure group, all subjects’ fetal stages must be after the GCF, born during 1963–1965. Considering there were a few years of differences in ages between the two groups, the non-exposure group’s physical examinations in this study were a few years later than those of the exposure group, so that age differences were balanced between the two groups. All subjects were given their informed consent for inclusion before they participated in the study. The study was conducted in accordance with the principles of the Declaration of Helsinki, and all protocols were approved by the Institute Ethics Committee of the First Hospital of Soochow University (ER0201601197).

All subjects in this study underwent cardiovascular examinations by hospital doctors. The examinations included an electrocardiogram (ECG), measurements of blood pressure, and heart rhythm. Six groups were divided according to the ECG test, including bradycardia, tachycardia, premature beat, pre-excitation syndrome, and premature beat and pre-excitation syndrome. For all suspected CVD cases, a panel of three experienced physicians, consisting of neurologists, cardiologists, and radiologists, reviewed the medical records. They defined common heart diseases on the basis of clinical symptoms, dynamic changes in cardiac enzymes and/or biomarker concentrations, and electrocardiogram results. Bradycardia is defined as a sinus rhythm is less than 60 beats per minute. Bradycardia on an ECG was shown as follows: (1) The first characteristic of sinus bradycardia on the ECG was the regular occurrence of a sinus p wave, but the frequency of the p wave was relatively small. The p wave in the I, II, AVF, V4-V6 leads was upright, and inverted in the aVR lead; (2) The PR interval was greater than 0.12 s; (3) The T wave amplitude was usually low and the QT interval was longer than usual. Tachycardia is defined as a sinus rhythm frequency was >100 beats/min. Tachycardia on an ECG was described as follows: (1) p Baud point: Generally, sinus tachycardia showed a normal p wave, but compared with the normal ECG, the p wave amplitude was slightly higher; (2) The PR interval was 0.12~0.20 s; (3) The PP interval may have mild irregularity (shortening or prolonging), but the difference between shortening and prolonging was not more than 0.12 s; (4) Normal QRS waveform; the PR interval and QT interval were shortened correspondingly. Myocardial ischemia showed T wave changes or ST segment changes, or occurred at the same time, and there were four forms of ECG expression: during a typical myocardial ischemic attack, the ST segment showed depression (horizontal or downward shift ≥0.1 MV) and/or T wave inversion; dome people may have persistent ST segment changes (horizontal or downward sloping type downward shift ≥0.05 MV) and/or T-wave inversion, negative positive bidirectional, or inversion; the coronary T wave may appear in coronary heart disease (inverted deep apex, symmetrical T waves in both limbs), these were also seen in patients with myocardial infarction; variant angina usually causes transient ST segment elevation and a towering T wave with a corresponding lead ST segment decrease. If the ST segment continued to rise, it indicated that myocardial infarction may occur. Blood pressure measurements were performed on the nondominant arm with two measurements separated by 3-min intervals after at least a 15-min rest. Blood pressure was divided into borderline hypertension and three stages of hypertension: Grade 1 hypertension was defined as systolic pressure (SP) 140–159 mmHg and diastolic pressure (DP) 90–99 mmHg; Grade 2 hypertension was defined as SP 160–179 mmHg and DP 100–109 mmHg. Grade 3 hypertension is known as SP ≥ 180 mmHg or DP ≥ 110 mmHg. This study collected data from patients with hypertension before starting the treatments for the both control and GCF exposure groups. A heartrate greater than 100 per minute is defined as tachycardia, and a rate less than 60 per minute is defined as bradycardia. The diagnosis of myocardial ischemia, premature beat, pre-excitation syndrome, myocardial ischemia, atrial fibrillation, and atrioventricular block was made by physicians. A blood sample from a blood vessel in the arm was collected in the morning after an overnight fast of at least 10 h. Total cholesterol, triglyceride, and blood glucose levels were measured on an autoanalyzer (the c16000 system, ARCHITECT ci16200 analyzer; Abbott Laboratories, Lake Bluff, IL, USA) in the hospital laboratory. Mean arterial pressure was calculated as (systolic blood pressure + 2 × diastolic blood pressure)/3. The formula for calculating body mass index (BMI) is: BMI = weight (kg) ÷ height^2^(m). Using 24 as the cutoff for BMI and 5.2 as the cutoff for total cholesterol, according to international standards, those exceeding the cutoffs are defined as having a high BMI or high cholesterol.

### 2.3. Statisticsh

All analyses were performed with IBM SPSS Statistics version 26 (IBM Corporation, Armonk, NY, USA). This was a cross-sectional study. Unless otherwise stated, variables are presented as mean ± SD and (%) for categorical variables. The student’s *t*-test was used for statistical comparisons. Differences in sex, blood pressure, cardiac indexes, as well as heart rhythm between the exposure and non-exposure groups were compared using Chi-square test with a 95% confidence interval. The classification analysis was carried out based on the relationship between basic indicators and metabolic indicators. Statistical significance was defined as *p* < 0.05.

## 3. Results

A total of 10,065 participants were recruited for the present analyses. There were 6128 subjects in the GCF exposure group (male: 3321, female: 2807), and 3937 in the control (non-exposure) group (male: 2103, female: 1834) (Figure 1).

There was no significant difference in the sex ratio (*p* = 0.445), age (52.97 ± 0.84 vs. 52.94 ± 0.86; *p* = 0.082), BMI (24.62 ± 2.97 vs. 24.53 ± 2.97; *p* = 0.268), blood triglyceride (1.78 ± 1.53 vs. 1.74 ± 1.55; *p* = 0.189), height (165.90 ± 7.53 vs. 166.10 ± 7.67; *p* = 0.359), blood glucose (5.43 ± 1.33 vs. 5.41 ± 1.16; *p* = 0.457) between the two groups. However, SP (128.3 ± 16.8 vs. 129.1 ± 17.3; *p* = 0.035), DP (80.11 ± 11.59 vs. 82.51 ± 12.13; *p* < 0.001), MAP (96.17 ± 12.60 vs. 98.03 ± 13.16; *p* < 0.001), total cholesterol (4.96 ± 0.87 vs. 5.00 ± 0.83; *p* = 0.017), and heart rate (75.91 ± 10.87 vs. 77.04 ± 11.08; *p* < 0.001) were higher in the exposure group than in the non-exposure group. In addition, the values for SP, DP, MAP, BMI, total cholesterol, height, and blood glucose were higher in the GCF exposure group than those of the non-exposure group (Table 1).

Logistic regression analysis showed that although there was no significant difference in Borderline hypertension (OR = 1.029, 95%CI: 0.921–1.150, *p* = 0.614) and Grade 1 hypertension (OR = 1.103, 95%CI: 0.992–1.226, *p* = 0.069) following the GCF, Grade 2 hypertension (OR = 1.724, 95%CI:1.441–2.064, *p* < 0.001) and Grade 3 hypertension (OR = 1.480, 95%CI: 1.050–2.086, *p* = 0.025) were affected by the GCF in the exposure group (Table 2).

ECG examination were performed on a total of 8999 subjects. Among them, there were 3573 in the control group and 5426 in the exposure group. Bradycardia (OR = 1.383, 95%CI: 1.154–1.657, *p* < 0.001), myocardial ischemia (OR = 1.301, 95%CI: 1.135–1.490, *p* < 0.001), atrial fibrillation (OR = 1.931, 95%CI: 1.033–3.610, *p* = 0.039), and atrioventricular block (OR = 1.333, 95%CI: 1.034–1.719, *p* = 0.027) were greater in the exposure group when compared to the control group. When comparing tachycardia (OR = 1.450, 95%CI: 0.946–2.222, *p* = 0.086), premature beat (OR = 1.360, 95%CI: 0.901–2.054, *p* = 0.142), pre-excitation syndrome (OR = 1.317, 95%CI: 0.241–7.195, *p* = 0.750), and VPB (OR = 1.230, 95%CI: 0.547–2.762, *p* = 0.616), there was no significant differences between the two groups (Table 2).

The results indicated possible influences for sex (male), total cholesterol (≥5.2 mmol/L), diabetes mellitus, metabolic syndrome, and hypertension with exposure to GCF (all P for interaction <0.05) in the development of CVDs. Except for BMI, all tested factors suggested a significant tendency toward Grade 2 hypertension. Sex (male), total cholesterol (≥5.2 mmol/L), and metabolic syndrome (Yes) all indicated a significantly increased risk for Grade 3 hypertension. The tendency between GCF exposure and CVDs was not significantly modified by female status, total cholesterol (<5.2 mmol/L), or hypertension (No) (*p*-values for those interactions were all >0.05) (Table 3). 

For cardiac auscultation, cardiac rhythm was divided into premature ventricular beats and other types of arrhythmias. Premature ventricular beats in the analysis consist of occasional premature beats, frequent premature beats, and ventricular bigeminy or trigeminy. Cardiac auscultation was performed on 3439 subjects in the non-exposed group and 5286 persons in the exposure group. Chi-square analysis showed no significant differences in premature ventricular beats (*p* = 0.616) and other arrhythmias (*p* = 0.096) (Table 2). Logistic regression analysis revealed no difference in premature ventricular beats (OR = 1.230, 95%CI: 0.547–2.762, *p* = 0.616) and other arrhythmias (OR = 0.487, 95%CI: 0.205–1.157, *p* = 0.096) (Table 2).

## 4. Discussion

Many studies have shown a link between undernutrition or famine during pregnancy and cardiovascular diseases in the offspring [10,11]. The present study revealed that exposure to the GCF during early developmental periods was closely associated with increased risks of multiple cardiovascular diseases, including Grade 2 and Grade 3 hypertension, myocardial ischemia, bradycardia, atrial fibrillation, and atrioventricular block, in the offspring after 50 years of age. The findings demonstrated that: (1) even more than 50 years after the GCF, the long-term influence of perinatal undernutrition still significantly impacted cardiovascular health; and (2) the long-term influence of GCF was accompanied by multiple heart disorders in the offspring at an early stage of life.

The role of environmental insults during fetal life in developing adult diseases has been extensively studied [12,13,14]. A prospective cohort related to the Dutch Famine suggested that exposure to the famine was associated with a threefold higher risk of coronary heart disease, atherosclerosis, dyslipidemia, glucose intolerance, and kidney diseases in the middle-aged offspring [12]. Previous studies on blood pressure in the offspring of women prenatally exposed to famine showed an increase in either systolic pressure or diastolic pressure, or both [15]. Our research team first demonstrated the long-term influence of the GCF on blood pressure in the offspring several years ago. After that, other groups also reported that the GCF affected cardiovascular functions and blood pressure [4]. Hypertension is defined by various stages in clinical practice according to increased levels of BP and DP. It would be interesting to determine if the GCF could influence different degrees of hypertension when the offspring were aging. This study was the first to investigate a possible relationship between the GCF and Grades 1–3 hypertension.

The present study showed both systolic and diastolic pressure as well as MAP were significantly higher in the aged offspring of the exposure group. Accumulative evidence has proven undernutrition during pregnancy to be a critical risk factor, leading to increased blood pressure. In our analysis of the effects of the GCF on levels of blood pressure, our results first showed the influence of famine on three stages of hypertension in the elderly, revealing increased risks in Grade 2 and Grade 3 hypertension in the exposure group. Clinically, Grade 2 and 3 hypertension are defined as SP ≥ 160–179 mmHg and DP ≥ 100–109 mmHg, respectively. It is well known that high blood pressure is closely linked to myocardial ischemia [16,17]. The present study also provided new evidence that, following prenatal exposure to the GCF, the incidence of myocardial ischemia was significantly increased in the offspring when they were between 50 and 60 years old. The finding of that severe undernutrition during perinatal periods could significantly influence critical grades of hypertension in humans offers new insights into understanding of etiology of Grade 2–3 hypertension in fetal origins. 

In the present study, the heart rate was higher in the offspring of the exposure group. Although analysis did not indicate statistical differences between the two groups in tachycardia, premature beat, and pre-excitation syndrome, the new finding showed that bradycardia, atrial fibrillation, and atrioventricular block were significantly increased in the aged offspring following GCF. Logistic regression analysis confirmed that prenatal exposure to GCF was a critical risk factor for those types of arrhythmias. To date, this is the first to demonstrate a close link between perinatal undernutrition and an increased risk of certain types of arrhythmias, including bradycardia, atrial fibrillation, and atrioventricular block, in the offspring exposed to GCF in their early stages of aging.

It is known that aging is accompanied by an increasing incidence of arrhythmia, such as bradycardia [18,19,20,21]. In an animal model, prenatally nutrient-restricted sheep showed increased pulse pressure and baroreflex sensitivity in response to angiotensin II was blunted, but tachycardia following a decrease in central blood pressure was potentiated [22], indicating that prenatal undernutrition may program long-term heart dysfunction. Undernutrition is a major cause of intrauterine growth restriction (IUGR) [23]. In rats, IUGR was significantly linked to ventricular premature beats and tachycardia associated with increased diastolic pressure in the offspring. However, investigations on the relationship between prenatal nutrition and arrhythmia from animal models to humans are needed in translational medicine. Thus, this study not only showed that heart rate was altered but also demonstrated increased risks for bradycardia, atrial fibrillation, and atrioventricular block in older people with perinatal undernutrition histories. Notably, although previous experiments in animal models and investigations in humans pointed out that prenatal undernutrition could significantly increase the risk of cardiovascular diseases such as hypertension and myocardial ischemia [24], none of those reports revealed prenatal insults-affected detailed diseases in cardiovascular systems, including Grade 2 and Grade 3 hypertension, bradycardia, atrial fibrillation, and atrioventricular block. One of the implications of this study’s novel finding is that individuals with a history of undernutrition during pregnancy should be given special attention, and patients could be advised on the early prevention of heart diseases, as healthy diets and physical activities, as well as other approaches, can be used in the early prevention before getting old.

Despite a number of studies over the last three decades that have demonstrated a close link between perinatal malnutrition and cardiovascular diseases [25,26], the underlying mechanisms have not been fully understood. Previous experimental studies showed that maternal dietary restriction resulted in a higher prevalence of hypertension, obesity, and diabetes mellitus in the offspring [24,27,28]. The present study observed the clustering of multi-morbidity following exposure to the GCF in the offspring when they were entering the old stage between 50–60 years of age. It is known that increased BMI, cholesterol, or glucose in the body are major contributors to the development of hypertension and arrhythmia [29,30,31]. This study demonstrated that incidence of abnormal total cholesterol levels was significantly higher in the GCF exposure offspring than the control group; we also found that all metabolism-related indexes included in the analysis, with the exception of BMI, including total cholesterol, diabetes mellitus, and metabolic syndrome, contributed to increased sensitivity in the development of Grade 2 hypertension by perinatal undernutrition. Logistic regression analysis showed greater risk of Grade 2 and Grade 3 hypertension by exposure to the GCF in offspring with higher cholesterol (>5.2 mmol/L) or the presence of either diabetes or metabolic syndrome. What is novel and interesting from those analyses is that if total cholesterol levels in the circulation were above the normal range or metabolic syndrome was present, perinatal undernutrition (GCF) significantly increased the risk of Grade 3 hypertension. Notably, this grade of hypertension is the most severe or malignant hypertension in clinical practice [32]. The finding of a close link among prenatal malnutrition, total cholesterol, metabolic syndrome, and Grade 3 hypertension provides new insight into the etiology and pathology of the development of severe hypertension. 

Although previous laboratory studies showed that malnutrition could induce arrhythmia in animal models [33,34], the present study first demonstrated that famine is a significant risk for developing bradycardia, atrial fibrillation, and atrioventricular block, not tachycardia, in humans. In the GCF exposure offspring, the chi-square test demonstrated that if subjects had an increased BMI, total cholesterol, blood pressure, or metabolic syndrome, they were more vulnerable in developing bradycardia. In addition, analysis indicated that an increase in total cholesterol in circulation could make the GCF-exposed offspring more sensitive to developing atrial fibrillation. These data provide new information on a possible link among perinatal insults, postnatal clinical testing indexes, and certain types of arrhythmias for clinical practice. 

The analysis also revealed that, regardless of normal BMI values, total cholesterol concentrations lower than 5.2 mmol/L, or a negative diagnosis of metabolic syndrome, prenatal exposure to the GCF in the male offspring was associated with a higher risk for the development of Grade 3 hypertension. For the female offspring, exposure to the GCF was a significant risk factor for inducing Grade 2 hypertension without association with normal or abnormal BMI indexes. That novel information strongly suggests there might be sex differences in CVDs in the offspring 50 years after exposure to the GCF. Compared to the female offspring, the male offspring seemed more vulnerable to the development of bradycardia and atrioventricular block after GCF. It is well known that sex hormones such as estrogen could be one of the protective mechanistic factors for female cardiovascular systems [35,36]. Whether the endocrine mechanisms were involved in the observed phenomenon in the present study requires further investigation.

The major strength of the present study is the large, well-characterized number of cohorts, which guaranteed a large number of hypertension and cardiovascular disease cases and limited the sampling bias. Importantly, we performed a comparison between subjects of the same age (52–55 years old), which provided much more convincing results in disclosing the effect of famine exposure on CVD risks. The other strengths of the present study include that birth dates were accurate and that physical examinations and data were performed and obtained by professional experts/doctors rather than personal self-reports. However, several limitations should be acknowledged in the current study. Firstly, birth weight was not available in the present study because there were no such medical records in China half century ago. Thus, we could not comment on the influence of birth weight on the increased risks of CVDs. Secondly, because of the cross-sectional design of the present study, which has its limitations, further investigations of the deep mechanisms underlying the GCF-affected CVDs still need to be conducted, probably in animal models. 

## 5. Conclusions

This study discovered that prenatal exposure to the GCF significantly increased the risk of cardiovascular diseases even more than 50 years after the famine and the suffering babies became old offspring, indicating the long-term impact of prenatal undernutrition playing important roles in the development of cardiovascular disorders in humans at an early stage of aging. The data analysis offered new information for further understanding the etiology and pathophysiology of cardiovascular outcomes following prenatal undernutrition, as well as expanding new knowledge for early prevention and treatment of cardiovascular diseases with developmental origins before reaching old age.

## Figures and Tables

**Figure 1 jpm-13-00163-f001:**
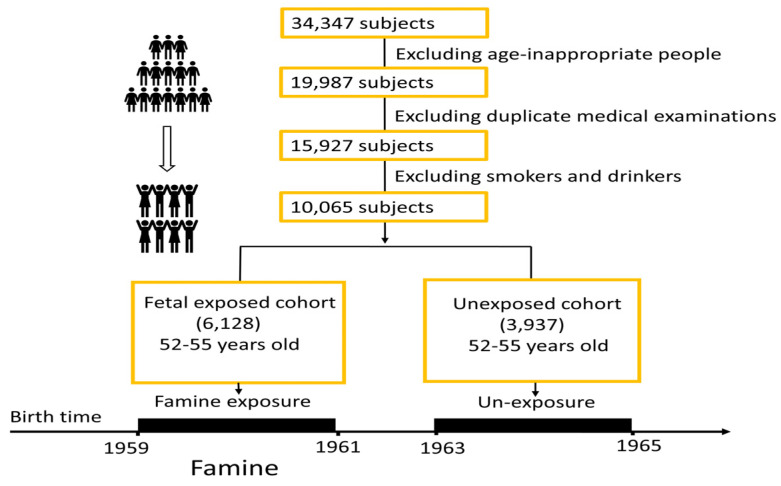
Flowchart of subjects.

**Table 1 jpm-13-00163-t001:** Cardiovascular and blood values between the exposure and non-exposure groups.

	Non-Exposure to GCF			Fetal Exposure to GCF			*p* Value
	Range	%	Mean ± SD	Range	%	Mean ± SD	
Sex (male%)	-	-	-	-	-	-	0.445
Male	-	53.42	-	-	54.19	-	-
Female	-	46.58	-	-	45.81	-	-
Age (years)	52–55	-	52.97 ± 0.84	52–55	-	52.94 ± 0.86	0.082
SP (mmHg)	80–192	-	128.3 ± 16.8	86–217	-	129.1 ± 17.3	0.035
DP (mmHg)	45–131	-	80.11 ± 11.59	48–138	-	82.51 ± 12.13	<0.001
MAP (mmHg)	56.67–155.00	-	96.17 ± 12.60	62.33–159.67	-	98.03 ± 13.16	<0.001
BMI (kg/m^2^)	15.72–35.01	-	24.62 ± 2.97	15.80–36.95	-	24.53 ± 2.97	0.268
<24	15.72–23.99	42.99	-	15.82–23.99	43.82	-	-
≥24	24.00–35.01	57.01	-	24.00–36.95	56.18	-	-
Total cholesterol (mmol/L)	2.22–9.42	-	4.96 ± 0.87	2.81–9.87	-	5.00 ± 0.83	0.017
<5.2 mmol/L	2.22–5.19	59.88	-	2.23–5.19	63.23	-	-
≥5.2 mmol/L	5.20–9.42	40.12	-	5.20–9.87	36.77	-	-
Triglyceride (mmol/L)	0.26–9.70	-	1.78 ± 1.53	0.60–9.43	-	1.74 ± 1.55	0.189
Height (cm)	138.6–188.4	-	165.90 ± 7.53	152.8–192.0	-	166.10 ± 7.67	0.359
Blood glucose (mmol/L)	3.23–20.67	-	5.43 ± 1.33	4.80–21.47	-	5.41 ± 1.16	0.457
Heart Rate (bpm)	46–150	-	75.91 ± 10.87	48–150	-	77.04 ± 11.08	<0.001

GCF, Great Chinese Famine; SP, systolic pressure; DP, diastolic pressure; MAP, mean arterial pressure; BMI, body mass index.

**Table 2 jpm-13-00163-t002:** The effects of the Great Chinese Famine on different stages of hypertension and cardiovascular diseases in electrocardiogram (ECG) or auscultation (AUSC).

	Non-Exposure (%)	Exposure (%)	OR	95%Cl	*p* Value
Borderline hypertension	20.48	19.62	1.029	0.921–1.150	0.614
Grade 1 hypertension	22.80	23.42	1.103	0.992–1.226	0.069
Grade 2 hypertension	5.19	8.33	1.724	1.441–2.064	<0.001
Grade 3 hypertension	1.37	1.88	1.480	1.050–2.086	0.025
Bradycardia (ECG)	5.18	7.02	1.383	1.154–1.657	<0.001
Tachycardia (ECG)	0.87	1.25	1.450	0.946–2.222	0.086
Premature beat (ECG)	0.95	1.29	1.360	0.901–2.054	0.142
Pre-excitation syndrome (ECG)	0.06	0.07	1.317	0.241–7.195	0.750
Myocardial ischemia (ECG)	9.91	12.51	1.301	1.135–1.490	<0.001
Atrial fibrillation (ECG)	0.36	0.76	1.931	1.033–3.610	0.039
Atrioventricular block (ECG)	2.52	3.61	1.333	1.034–1.719	0.027
VPB (AUSC)	0.26	0.32	1.230	0.547–2.762	0.616
Other arrhythmias (AUSC)	0.35	0.17	0.487	0.205–1.157	0.096

VPB, ventricular premature beat; ECG, electrocardiogram; AUSC, auscultation of the heart. (ECG) Non-exposure: *n* = 3573, exposure: *n* = 5426. (AUSC) Non-exposure: *n* = 3439, Exposure: *n* = 5286.

**Table 3 jpm-13-00163-t003:** The influence for certain factors with exposure to GCF in the development of the CVDs.

*p*-Value	Grade 2 Hypertension	Grade 3 Hypertension	Bradycardia	Myocardial Ischemia	Atrial Fibrillation	Atrioventricular Block
Sex						
Men	<0.001	0.041	0.013	0.012	0.988	0.015
Women	<0.010	0.372	0.606	0.582	0.910	0.361
BMI						
<24	0.435	0.256	0.151	0.443	0.728	0.738
≥24	0.325	0.173	<0.005	<0.001	0.531	0.736
Total cholesterol						
<5.2 mmol/L	<0.001	0.597	0.247	0.858	0.838	0.211
≥5.2 mmol/L	<0.001	0.011	0.023	<0.001	0.473	0.026
Diabetes Mellitus						
Yes	<0.001	0.187	0.338	0.049	0.850	0.653
No	<0.001	0.093	0.415	0.019	0.853	0.505
Metabolic syndrome						
Yes	<0.001	<0.005	<0.001	0.011	0.687	0.506
No	<0.001	0.183	0.036	0.094	0.804	0.775
Hypertension						
Yes	-	-	0.035	0.026	0.449	0.320
No	-	-	0.927	0.969	0.550	0.546

BMI, Body Mass Index.

## Data Availability

Supporting data can be obtained from the corresponding author.

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
