# Peer review of "Long-Term Impact of the Great Chinese Famine on the Risks of Specific Arrhythmias and Severe Hypertension in the Offspring at an Early Stage of Aging"

_jpm, 2023, doi:10.3390/jpm13020163_

Round 1

Reviewer 1 Report

Well written manuscript on very interesting topic. However, there are several methodological issues:

1. Major limitation is that there is no data available on birth weight and that presumed exposure to famine was based purely on date of birth. Please provide relevant data regarding the extent of famine in geographical area where the study was undertaken.

2. Please provide details from where exactly all the patients were included in the study as the authors provide that patients were mainly included from one hospital without exact percentage. Additionally, please provide time frame when the data was collected (ie 2018 -2021)

3. Please provide definitions for bradycardia, tachycardia and myocardial ischemia on ECG.

4. Please separate category AV block in more clinically meaningful categories (AV block grade I, II and III).

5. Please clarify if the patients diagnosed with hypertension were on treatment, and if so, provide details about it. 

Author Response

Response to Reviewer 1Comments

Point 1: Well written manuscript on very interesting topic. However, there are several methodological issues.

Response 1:  Thanks for the comments.

Point 2: Major limitation is that there is no data available on birth weight and that presumed exposure to famine was based purely on date of birth. Please provide relevant data regarding the extent of famine in geographical area where the study was undertaken.

Response 2: A good question. However, Chinese Great Famine (CGF) occurred 50 years ago when whole country was under terrible economic conditions. At that period, Chinses medical service of OB & GYN was very limited and lacked properly trained medical professinals and systems. In Fact, there were no birth weight records at that period in China. However, since three-year famine affected whole country and whole populations (it was reported that millions died from hungery), it is reasonable many low birth weight happened during CGF. Regarding geographical area Suzhou region where the study was underteken, the whole region and whole local people were severely affected by CGF too.

Point 3: Please provide details from where exactly all the patients were included in the study as the authors provide that patients were mainly included from one hospital without exact percentage. Additionally, please provide time frame when the data was collected (ie 2018 -2021)

Response 3: A very reasonable question. The patients and their data were from First Hospital of Soochow University that is the largest hospital in the area and patients were from whole region in Suzhou area. For exposed group, the data were collected from 2013 year (collecting people borning from 1960-1961) and 2014 year (collecting people borning from 1959-1961). For un-exposed group, the data were collected from 2017 year (collecting people borning from 1963-1965) and 2018 year ( collecting people borning from 1963-1965).

Point 4: Please provide definitions for bradycardia, tachycardia and myocardial ischemia on ECG.

Response 4:

(1) Bradycardia means adult sinus rhythm is less than 60 beats per minute and ECG expressed as below:1. The first characteristic of sinus bradycardia electrocardiogram is the regular occurrence of sinus P wave, but the frequency of P wave is relatively small, P wave in the â… , â…¡, AVF, V4 to V6 lead upright, in aVR lead inverted. 2. P-R interval greater than 0.12 seconds. 3. The T wave amplitude is usually low and the Q-T interval is longer than usual. 4. U wave is sometimes prominent.

(2) Tachycardia means adult sinus rhythm frequency > 100 beats/min, ECG expressed as below: 1. p Baud point: Generally speaking, sinus tachycardia has normal p wave, but compared with normal ECG, p wave amplitude can be slightly higher; 2. P-R interval: 0.12~0.20 seconds; 3. P-P interval: the P-P interval may have mild irregularity (shortening or prolonging), but the difference between shortening and prolonging is not more than 0.12 seconds; 4. qrs waveform: Normal qrs waveform 5. The PR interval and QT interval are shortened correspondingly, sometimes accompanied by secondary mild ST segment depression and T wave amplitude reduction.

(3) myocardial ischemia shows T wave changes or ST segment changes, or occurs at the same time and here are four forms of expression: During typical myocardial ischemic attack, ST segment depression (horizontal or downward shift ≥ 0.1MV) and/or T wave inversion; Some people may have persistent ST segment changes (horizontal or downward sloping type downward shift ≥ 0.05MV) and/or T-wave inversion, negative positive bidirectional, inversion; Coronary T wave may appear in coronary heart disease (inverted deep apex, symmetrical T wave in both limbs), also seen in patients with myocardial infarction; Variant angina usually causes transient ST segment elevation+towering T wave and corresponding lead ST segment decrease. If ST segment continues to rise, it indicates that myocardial infarction may occur.

Point 5: Please separate category AV block in more clinically meaningful categories (AV block grade I, II and III).

Response 5: It is good idea to separate category AV block. However, for most cases in this study, we only got initial diagnosis of AV block without getting their further testing results in difining AV block grade I, II, and III.

Point 6: Please clarify if the patients diagnosed with hypertension were on treatment, and if so, provide details about it.

Response 6: A very good question too. This study collected the data of the patients with hypertension before starting the treatments for the both control and CGF exposure groups.

Reviewer 2 Report

In the manuscript titled, “Long-term Impact of Chinese Great Famine on Risks of Specific Arrythmias and Severe Hypertension in the Offspring at Early Stage of Aging”, Qiutong Zheng and colleagues were interested in exploring the long-term cardiovascular and arrhythmic effects on offspring who spent their fetal life during the Chinese Great Famine (CGF). They found that perinatal undernutrition during CGF was associated with increased risk of hypertension, myocardial infarction, bradycardia, and atrial fibrillation among other. I found the manuscript to be relatively well-explained and the figure and tables easy to comprehend but do harbor some concerns.

Major concerns:

1.      Was birth period (1959-1961) the only criteria to define perinatal undernutrition? I think there needs to be a distinction that needs to be made regarding the exposure group: is it perinatal undernutrition (for which I would suggest certain growth parameters among offspring such as small for gestational age) or prenatal exposure to CGF (being born during this time period)?

2.      The study is based on observational measures and outcomes and does not explore the underlying mechanisms at a person-based level. Therefore, I don’t think it is appropriate to make claims regarding information regarding personalized medicine.

3.      In lines 156-163, the authors comment on interactions between sex and other covariates and CGF. What does interaction refer to? This does not appear to be interaction effects and constructed regression analyses, none of which is mentioned in the Statistics section in Materials and Methods. Please elaborate and replace with proper terminology.

Minor concerns:

1.      There are some grammar related issues throughout the manuscript—please address them. Also, similar to your previous publication (PLOS One, 2017), I would suggest using the Great Chinese Famine (GCF).

2.      In lines 125-134, it appears the results are mentioned twice regarding Table 1. Please address this.

3.      The Discussion section can be reworded and shortened, and statistical analyses do not need mention here.

Author Response

Response to Reviewer 2Comments

Point 1: In the manuscript titled, “Long-term Impact of Chinese Great Famine on Risks of Specific Arrythmias and Severe Hypertension in the Offspring at Early Stage of Aging”, Qiutong Zheng and colleagues were interested in exploring the long-term cardiovascular and arrhythmic effects on offspring who spent their fetal life during the Chinese Great Famine (CGF). They found that perinatal undernutrition during CGF was associated with increased risk of hypertension, myocardial infarction, bradycardia, and atrial fibrillation among other. I found the manuscript to be relatively well-explained and the figure and tables easy to comprehend but do harbor some concerns.

Response 1:  Thanks for the comments.

Point 2: Was birth period (1959-1961) the only criteria to define perinatal undernutrition? I think there needs to be a distinction that needs to be made regarding the exposure group: is it perinatal undernutrition (for which I would suggest certain growth parameters among offspring such as small for gestational age) or prenatal exposure to CGF (being born during this time period)?

Response 2:  Excellent question and comment. Chinese Great Famine (CGF) occurred 50 years ago when whole country was under terrible economic conditions and food shortage. At that period, Chinses medical service of OB & GYN was very limited and lacked properly trained medical professinals and systems. In Fact, there were no birth weight records at that period in China. However, since three-year famine affected whole country and whole populations (it was reported that millions died from hungery, and almost all photos taken during CGF showed severe under-nutrition men, women, and children), it is reasonable many low birth weight happened during CGF.

Point 3: The study is based on observational measures and outcomes and does not explore the underlying mechanisms at a person-based level. Therefore, I don’t think it is appropriate to make claims regarding information regarding personalized medicine.

Response 3:  Based on the comment, we rewrote the rlated sentences.

Point 4: In lines 156-163, the authors comment on interactions between sex and other covariates and CGF. What does interaction refer to? This does not appear to be interaction effects and constructed regression analyses, none of which is mentioned in the Statistics section in Materials and Methods. Please elaborate and replace with proper terminology.

Response 4:  Very nice question. Actually our analysis indicates an interaction trendency. After consulting Dr. Prof. Yi Ding who is an expert in analysis those kinds of data, we made changes and revision accordingly. For example, we changed “showed interaction effects” into “indicated possible tendency”.

Minor concerns:

Point 5: There are some grammar related issues throughout the manuscript—please address them. Also, similar to your previous publication (PLOS One, 2017), I would suggest using the Great Chinese Famine (GCF).

Response 5: Thanks. We did a throughout check of grammar, and changed the words into Great Chinese Famine (GCF).

Point 6: In lines 125-134, it appears the results are mentioned twice regarding Table 1. Please address this.

Response 6: We made corrections accordingly.

Point 7: The Discussion section can be reworded and shortened, and statistical analyses do not need mention here.

Response 7: Based on the comments, we shortemed the Discussion.

Reviewer 3 Report

Interesting paper on the long-term impact of the Great Chinese Famine, which occurred 50 years ago, on the risks of specific arrhythmias and severe hypertension in exposed individuals. Its objective is to determine the long-term impact of perinatal malnutrition on hypertension and arrhythmias in now-exposed elderly people.

Methodology and statistics appropriate to the purpose of the study.

Discussion and conclusion according to the results obtained.

Author Response

Response to Reviewer 3Comments

Response 1: Interesting paper on the long-term impact of the Great Chinese Famine, which occurred 50 years ago, on the risks of specific arrhythmias and severe hypertension in exposed individuals. Its objective is to determine the long-term impact of perinatal malnutrition on hypertension and arrhythmias in now-exposed elderly people.

Methodology and statistics appropriate to the purpose of the study.

Discussion and conclusion according to the results obtained.

Response 1: Thanks for the comments.

Round 2

Reviewer 1 Report

Please include your comments into the text of the manuscript, as this is the point of revision. No further comments

Author Response

Response to Reviewer 1Comments

Point 1: Please include your comments into the text of the manuscript, as this is the point of revision. No further comments

Response 1:  Agree. We did revision as suggested.